# Ammonium hydroxide treatment of Aβ produces an aggregate free solution suitable for biophysical and cell culture characterization

Timothy M. Ryan[1], Joanne Caine[2], Haydyn D.T. Mertens[3], Nigel Kirby[3], Julie Nigro[2], Kerry Breheney[2], Lynne J. Waddington[2], Victor A. Streltsov[2], Cyril Curtain[1], Colin L. Masters[1] and Blaine R. Roberts[1]

[1] The Florey Institute of Neuroscience and Mental Health, University of Melbourne, Parkville, Victoria, Australia
[2] Materials Science and Engineering, Preventative Health Flagship, Commonwealth Scientific and Industrial Research Organization, Parkville, Victoria, Australia
[3] SAXS/WAXS Beamline, Australian Synchrotron, Clayton, Victoria, Australia

## ABSTRACT

Alzheimer's disease is the leading cause of dementia in the elderly. Pathologically it is characterized by the presence of amyloid plaques and neuronal loss within the brain tissue of affected individuals. It is now widely hypothesised that fibrillar structures represent an inert structure. Biophysical and toxicity assays attempting to characterize the formation of both the fibrillar and the intermediate oligomeric structures of Aβ typically involves preparing samples which are largely monomeric; the most common method by which this is achieved is to use the fluorinated organic solvent 1,1,1,3,3,3-hexafluoro-2-propanol (HFIP). Recent evidence has suggested that this method is not 100% effective in producing an aggregate free solution. We show, using dynamic light scattering, size exclusion chromatography and small angle X-ray scattering that this is indeed the case, with HFIP pretreated Aβ peptide solutions displaying an increased proportion of oligomeric and aggregated material and an increased propensity to aggregate. Furthermore we show that an alternative technique, involving treatment with strong alkali results in a much more homogenous solution that is largely monomeric. These techniques for solubilising and controlling the oligomeric state of Aβ are valuable starting points for future biophysical and toxicity assays.

## INTRODUCTION

Alzheimer's disease is the leading cause of dementia in the elderly human population. This disease is characterised by neurofibrillary tangles, neuronal death, synapse dysfunction and formation of extracellular amyloid aggregates, ultimately leading to cognitive decline and dementia. Amyloid formation by Aβ peptides is characteristic of the disease, forming the

Corresponding author
Blaine R. Roberts,
blaine.roberts@florey.edu.au

major constituent of the plaques that are symptomatic of Alzheimer's disease (*Glenner & Wong, 1984*; *Masters et al., 1985*). However, the formation of these fibrillar structures is not well correlated with disease severity (*McLean et al., 1999*), indicating that fibrils represent a relatively non-toxic endpoint in a toxic aggregation pathway (*McLean et al., 1999*). To reconcile these observations several theories suggesting the presence of soluble, small, oligomeric species of A$\beta$ have been developed (*Kirkitadze, Bitan & Teplow, 2002*).

There are a number methods for generating small toxic oligomers *in vitro* (for examples see (*Sahoo et al., 2009*; *Yu et al., 2009*; *Ahmed et al., 2010*; *Ryan et al., 2010*)), but typically most of these preparations involve the preliminary generation of a stock of A$\beta$ which is largely monomeric (*Stine et al., 2003*). Generally this is a critical requirement for many kinetic studies, as solutions which contain a large number of aggregates will have adverse impacts on understanding how an unfolded form of A$\beta$ self-associates into a small nucleus which then elongates into fibrillar structures (*Stine et al., 2003*; *Teplow, 2006*). Unfortunately, this is not a simple process of resuspending lyophilised peptide in the desired buffer, as the hydrophobic nature of the peptide makes the initial dissolution very difficult (*Teplow, 2006*). Furthermore A$\beta$ appears to have a structural memory, reforming the conformations that it had adopted prior to lyophilisation (*Stine et al., 2003*).

To avoid these issues many researchers resort to a variety of solubilisation agents, including strong alkali and acids (*Zagorski et al., 1999*; *Fezoui et al., 2000*; *Teplow, 2006*), dimethylsulfoxide (DMSO) (*Shen & Murphy, 1995*; *Lambert et al., 2001*; *Broersen et al., 2011*), chaotropic salts and fluorinated alcohols, such as trifluroethanol (*Zagorski & Barrow, 1992*) and hexafluoroisopropanol (HFIP) (*Stine et al., 2003*; *Broersen et al., 2011*). The most commonly adopted approach is to use HFIP (thought to induce alpha helical conformations, produce monomeric solutions and remove the conformational memory from the peptide) followed by removal of HFIP and resuspension using an aqueous compatible resuspension solvent such as DMSO (*Walsh et al., 1997*; *Stine et al., 2003*; *Williams, Day & Serpell, 2010*). Recently, the use of HFIP has been suggested by a number of groups to cause heterogeneity in the starting peptide self-association state (*Nichols et al., 2005a*; *Nichols et al., 2005b*; *Pachahara et al., 2012*), which may cause enhanced aggregation.

We show using a variety of methods, including thioflavin T (ThT) binding, Small Angle X-ray scattering (SAXS), dynamic light scattering (DLS) and size exclusion chromatography measurements, that HFIP pretreatment affects the aggregation state and kinetics of amyloid formation of the resuspended peptide. We also show that the use of ammonium hydroxide, as an initial treatment, greatly aids solubility and produces a relatively less aggregated and seed free peptide solution, which is important for understanding the process of A$\beta$ self-association and hence neuronal cell toxicity.

## MATERIALS

Human A$\beta_{1-42}$ (from this point referred to as A$\beta_{42}$) was synthesised and purified by Dr. James I. Elliott at Yale University (New Haven, CT) using tBOC chemistry with DCC and HOBT coupling reagents. The correct unmodified sequence was verified by mass

spectrometry. 1,1,1,3,3,3-hexafluoro-2-propanol (HFIP) was obtained from Oakwood Products Inc. (West Columbia, SC), analytical grade ammonium hydroxide (NH$_4$OH) (28% v/v), trifluoroacetic acid (TFA, 99%) and staurosporine from *Streptomyces* sp. were from Sigma Chemical Co (St Louis, MO). Poly-L-lysine coated 96-well plates were from BD (Franklin Lakes, NJ Cat. No. 356516). The F12K media, horse serum, penicillin, streptomycin, amphotericin B and trypsin/EDTA were from Invitrogen (Grand Island, NY). Foetal bovine serum (FBS) was from SAFC Biosciences (Lenexa, KS). Human recombinant $\beta$-nerve growth factor (NGF) was purchased from Sino Biological Inc. (Beijing, P.R. China). The cell counting kit-8 (CCK-8) reagent was from Dojindo Molecular Technologies, Inc. (Kumamoto, Japan). All reagents otherwise not indicated were of analytical grade.

## METHODS

### A$\beta$ HFIP and NH$_4$OH pretreatment

Synthetic human A$\beta_{42}$ peptide was prepared in two separate ways. Firstly the A$\beta_{42}$ lyophilised powder was dissolved in TFA and dried under a nitrogen stream. The remaining film was dissolved in 100% HFIP to a concentration of 1 mg/ml, sonicated 5 min in a bath sonicator and dried under a nitrogen stream. The HFIP treatment was repeated twice more and on final dissolving the peptide was dispensed into microcentrifuge tubes. After drying under a nitrogen stream, the peptide was further dried under vacuum for 1–2 h to give a clear film.

The second method was to take the original peptide and dissolve it in 10% (w/v) NH$_4$OH at 0.5 mg/ml. The peptide was incubated for 10 min at room temperature followed by sonication (5 min) and then dispensed (0.5 ml) into microfuge tubes. The NH$_4$OH was removed by lyophilisation to yield a salt free fluffy white peptide. All aliquots from both methods were then stored at −80°C. Immediately prior to use, the HFIP- and NH$_4$OH-treated and untreated A$\beta_{42}$ were dissolved in 60 mM NaOH and the concentration determined by absorbance at 214 and 280 nm using extinction coefficients of 76848 M$^{-1}$ cm$^{-1}$ or 1490 M$^{-1}$ cm$^{-1}$, respectively. In all cases the resuspended peptide was analysed by mass spectrometry and the peptide found to be unmodified with no trace of HFIP or NH$_4$OH.

### Thioflavin T assay

The ThT assays were conducted as described previously (*McColl et al., 2009*; *Ryan et al., 2012*), Briefly, A$\beta_{42}$ reconstituted in 60 mM NaOH to a stock concentration ∼200 μM was diluted to a final concentration of 5 μM in phosphate buffered saline (PBS, 136.89 mM NaCl, 2.68 mM KCl, 6.39 mM Na$_2$PO$_4$, 1.47 mM KH$_2$PO$_4$) pH 7.4 containing 30 μM ThT. Vehicle controls of 60 mM NaOH were used to confirm that the typical dilution of between 1 in 50 to 1 in 200 did not significantly alter the pH of the PBS solution. The samples were dispensed (200 μl) into a 96 well clear bottom plate and placed in a Flexstation 3 plate reader (Molecular Devices, Sunnyvale, CA), incubated at 37°C with agitation every 7 min. The ThT fluorescence was measured every 7 min for 24 h using excitation and emission wavelengths of 440 nm and 485 nm, respectively.

## Transmission Electron Microscopy (TEM)

Carbon-coated 300-mesh copper grids were glow-discharged in nitrogen to render the carbon film hydrophilic. Samples were gently agitated, before pipetting (4 μl) onto the grids. After 30 s adsorption time, the excess was drawn off using Whatman 541 filter paper. The grids were stained with 2% w/v potassium phosphotungstate, pH 7.2, for 10 s. Grids were air-dried before examination. The samples were examined using a Tecnai 12 Transmission Electron Microscope (FEI, Eindhoven, The Netherlands) at an operating voltage of 120 KV. Images were recorded using a Megaview III CCD camera and AnalySIS camera control software (Olympus).

## Dynamic light scattering

For dynamic light scattering (DLS) experiments the $A\beta_{42}$ dissolved in 60 mM NaOH (Stock concentration 500 μM) was diluted into 50 mM Tris pH 8.0 buffer containing 10 mM EDTA to a final concentration of 100 μM. This buffer was chosen to minimise aggregation as $A\beta$ aggregates at a slower rate at this pH and EDTA was included to prevent metal induced aggregation (*Huang et al., 2004*). Thus, the aggregation state of the resuspended material was not expected to change significantly during the time taken to prepare the peptide for DLS measurements. Vehicle controls were prepared at the concentrations used in the $A\beta_{42}$ experiments. The pH of all reactions were checked to be pH 8.0. All reaction mixtures were filtered through a 0.2 μm syringe filter before the concentration of the sample was measured to determine the degree of peptide loss during filtration. In all cases the concentration of peptide changed less than 5%. Subsequently, 20 μl of the filtrate was loaded in quadruplicate into a Greiner 384-well low volume glass bottom plate. The plate was centrifuged (5 min, $1000 \times g$) to pellet any remaining larger insoluble material prior to measurement. A total time of 20 min elapsed between the addition of the $A\beta_{42}$ to the buffer and the acquiring of data from the plate.

The DLS measurements were taken at 25°C using a Dyna Pro Nano Star plate reader at the Bio21 Collaborative Crystallisation Centre at laser wavelength 830 nm, with temperature control capacity and auto-attenuation function. To ensure accurate readings, the final distribution was taken from an accumulation of 50 individual, 5 s DLS collections. Hydrodynamic radius ($R_h$) was calculated from the diffusion coefficient by the Stokes–Einstein relationship.

## Size Exclusion Chromatography (SEC)

For SEC measurements the $A\beta_{42}$ stock in 60 mM NaOH was diluted into PBS to a final concentration of 200 μM and incubated at 25°C for 20 min. The samples were centrifuged (5 min, $16,500 \times g$) immediately before loading onto a S200 Superdex column (GE, 10/300 GL). The column was eluted with 50 mM Tris pH 8.0 at a flow rate of 0.5 ml/min for 1.5 column volumes. The elution of proteins was measured using a flow cell absorbance detector set at 280 nm. Molecular weight was estimated by comparison to the SEC protein calibrants (GE Healthcare).

## Small-angle X-ray scattering (SAXS) measurements

A slightly modified approach for solubilising the treated $A\beta_{42}$ was used for SAXS measurements, whereby the lyophilised peptide (either treated with HFIP or $NH_4OH$, and typically in 3–4 mg aliquots) was taken up in 20 µl of 60 mM NaOH, which was immediately diluted to 13 mM NaOH with distilled water. This sample was sonicated for 5 min in a bath sonicator to ensure the peptide entered the solution at the lower concentration of NaOH. The sample was then neutralised with 10X PBS, resulting in a solution with a pH of 7.4 and centrifuged at $16,500 \times$ g in a benchtop microcentrifuge for 10 min. The 214 nm absorbance of the supernatant was measured to determine concentration using the above quoted extinction coefficient. Typically this process resulted in stock solutions of approximately 5–7 mg/ml (1300–1500 µM). No differences in the kinetics of fibril formation, as measured by ThT fluorescence, were observed for direct dilution from 60 mM NaOH or 13 mM NaOH.

SAXS measurements were acquired at the SAXS/WAXS beamline of the Australian Synchrotron (Clayton, Victoria, Australia). $A\beta_{42}$ solutions, made using untreated, HFIP treated and $NH_4OH$ treated lyophilised peptide, solubilised using the above described NaOH solubilisation protocol (1 mg/ml, 221 µM) were analysed at a camera length of 3.3 m corresponding to a range of momentum transfer $0.005 \leq q \leq 0.35$ Å$^{-1}$ (where $q = 4\pi \sin\theta/\lambda$, $2\theta$ is the scattering angle and $\lambda$ is the X-ray wavelength: 1.03 Å at 12 KeV), using a Pilatus 1 M camera (Dectris, Switzerland). Data were normalised using an integrated beamstop and intensities put on an absolute scale using distilled water as a standard. To limit radiation damage, 50 µl samples/buffers were flowed through a 1.5 mm quartz capillary and 10 frames of 1 s exposure collected. The individual frames were compared for agreement before being averaged with the scatterBrain IDL program, downloaded from the Australian synchrotron SAXS/WAXS beam line website (http://www.synchrotron.org.au/).

SAXS data were analysed using Primus (*Konarev et al., 2003*) from the ATSAS (*Petoukhov et al., 2007*; *Petoukhov et al., 2012*) software suite (www.embl-hamburg.de/biosaxs/software.html), which was used to plot the data and create Kratky plots. A Kratky plot, is a plot of Intensity (I) multiplied by the square of the scattering vector, *s*, (defined as equal to $4\pi \sin(\theta)/\lambda$, where $2\theta$ is the total scattering angle and $\lambda$ is the wavelength of the X-ray) against *s* (*Doniach, 2001*).

## $A\beta_{1-42}$ toxicity in cell culture

Rat pheochromocytoma (PC12) cells were maintained and passaged every 3–4 days in PC12 media (F12K media containing 15% (v/v) horse serum, 2.5% (v/v) fetal bovine serum (FBS), 100 U/ml penicillin G sodium, 100 µg/ml streptomycin sulfate and 0.25 µg/ml amphotericin B. The PC12 cells (passage 5–15) were trypsinised (0.05% (w/v) trypsin/0.48 mM EDTA in PBS) and harvested in the maintenance media. The cells were centrifuged ($400 \times$ g, 5 min) and the cell pellet was resuspended in F12K media containing 0.5% (v/v) FBS, 100 U/ml penicillin G sodium, 100 µg/ml streptomycin sulfate and 0.25 µg/ml amphotericin B). The centrifugation step was repeated and the cell

pellet was resuspended in the media containing 0.5% (v/v) FBS with the addition of 100 ng/ml NGF. The cells were seeded at $5 \times 10^3$ cells/well (100 μl/well) into the inner 60 wells of a 96-well plate coated with poly-L-lysine. The outer 36 wells were used as a moat and were filled with sterile water (100 μl/well). The cultures were maintained at 37°C with 5% $CO_2$ in a humidified incubator (Heraeus Hera cell 150, Kendro Laboratory Products, Langenselbold, Germany) and allowed to differentiate for 96 h before treating with $A\beta_{42}$.

The HFIP- or $NH_4OH$-treated peptide were prepared as described for the SAXS measurements, except 0.5 mg of $A\beta$ was reconstituted using 60 mM NaOH, sterile water and 10 × Dulbecco's PBS consisting of 1368.9 mM NaCl, 26.8 mM KCl, 63.9 mM $Na_2PO_4$, 14.7 mM $KH_2PO_4$, 9.0 mM $CaCl_2$ and 4.9 mM $MgCl_2$ (*Dulbecco & Vogt, 1954*) instead of the standard laboratory reagents. The resulting $A\beta$ peptide was assessed for concentration as described for the SAXS measurements. This process resulted in $A\beta$ stock solutions of 180–250 μM. The concentrated peptide was diluted into the cell culture media A (F12K, 0.5% FCS, 100 ng/ml NGF, 100 U/ml penicillin G sodium, 100 μg/ml streptomycin sulfate and 0.25 μg/ml amphotericin B) at no more than 1/10 so as not to affect the composition of the culture media. The vehicle was prepared in the culture media using the same dilution as the peptide. The media/vehicle was used to make subsequent dilutions of the peptide (10–20 μM). Cultures were also treated with the toxin control, staurosporine (1 μM).

The cultures were incubated and assessed for viability after 48, 72 and 96 h using the CCK-8 reagent. Five minutes prior to the assay, a second toxin control (0.1% (v/v) Triton X-100) was added to the designated cells. The viability assay was done by aspirating the treatment media and replacing with cell culture media A containing 10% of the CCK-8 reagent (100 μl/well). The cultures were incubated for 2 h at 37°C and then measured for absorbance changes on the LabSystems Multiskan MS multiplate reader (ThermoFisher Scientific, Scoresby, Australia) using a 405 nm filter. Wells containing CCK-8 reagent and no cells were used to measure the background absorbance of the CCK-8/media and the values were subtracted from the test data. The absorbance readings for the untreated cells were normalised to 100% to adjust for variation between control values of individual experiments and the test data were expressed as a proportion of the untreated cells. The normalised data represent the mean and standard error of the mean (SEM) from 2–4 experiments. The normalised comparisons and the results were taken to be significant when the *p* value was below 0.05. The ANOVA and comparisons were done in a Microsoft® Office Excel spreadsheet.

## RESULTS

### Effect of solubilisation pretreatment method on A*β* amyloid formation

The effect of HFIP treatment on amyloid formation by $A\beta_{42}$ peptides was investigated using a continuous ThT assay. ThT fluorescence was measured for solutions containing untreated, HFIP or $NH_4OH$ pretreated $A\beta_{42}$ (Fig. 1A) and resuspended using the standard NaOH protocol. The ThT fluorescence timecourse of untreated $A\beta_{42}$ was variable with an average $t_{1/2}$ of approximately $14.9 \pm 2.6$ h (Fig. 1A). Pretreatment with HFIP decreased the

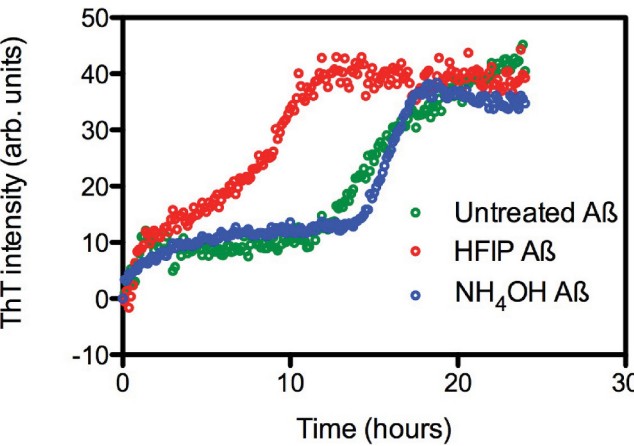

**Figure 1 HFIP treatment significantly enhances the rate of fibril formation.** (A) Analysis of HFIP, $NH_4OH$ effect on the aggregation kinetics of $A\beta_{42}$ by ThT fluorescence time course. $A\beta_{42}$ (5 μM) treated with HFIP (red circles, $t1/2 \sim 9 \pm 1.9$ h), $NH_4OH$ (blue circles, $t1/2 \sim 16.2 \pm 1.3$ h), or not treated at all (green circles, $t1/2 \sim 15 \pm 2.6$ h) prior to suspension into PBS.

$t1/2$ by 6 h to approximately $9 \pm 1.9$ h, while $NH_4OH$ pretreatment increased the average $t1/2$ to approximately $16.2 \pm 1.3$ h. In all cases we observe an initial increase in the ThT fluorescence intensity over the first hour, equivalent to approximately 10 arbitrary units, which is within the background of the timecourse assay and does not differ with $A\beta_{42}$ treatment. This increase could be attributed to a number of early events, ranging from temperature equilibration in the instrument through to the formation of cross-$\beta$ sheets structures and early oligomers.

### Aggregation state of A$\beta$ after dissolution.

Due to the observed increase in the rate of fibril formation that occurs with HFIP pretreatment, we hypothesised that this solvent was not completely removing the oligomers, but was in fact allowing amyloid seeds to persist through the complete removal of HFIP and thereby accelerate aggregation and fibril formation. To investigate the starting aggregation state of $A\beta_{42}$ in solution we used dynamic light scattering, size exclusion chromatography and small angle X-ray scattering. DLS was used to investigate qualitative differences in the preparation techniques. DLS provides an approximate hydrodynamic radius based on the ability of a solution to scatter light and is sensitive to the presence of aggregates. While hydrodynamic radius is typically applied to spherical particles, in this case the approximate radius is sufficient to indicate a qualitative difference between the two treatments. The DLS measurements on average demonstrated that both the HFIP and $NH_4OH$ treated $A\beta_{42}$ exist as a monomodal polydisperse solution (Fig. 2A) however, the spread of data for both samples was quite wide, with the extremes of low and high distributions shown in the supplementary data (Fig. S1). For the HFIP preparations we found a large degree of variability in the average distribution of five separate experiments with polydispersity factors ranging from 10%–52% and radii ranging from 3–100 nm (Fig. 2A, red line). Given this variability, the most commonly observed average radius was

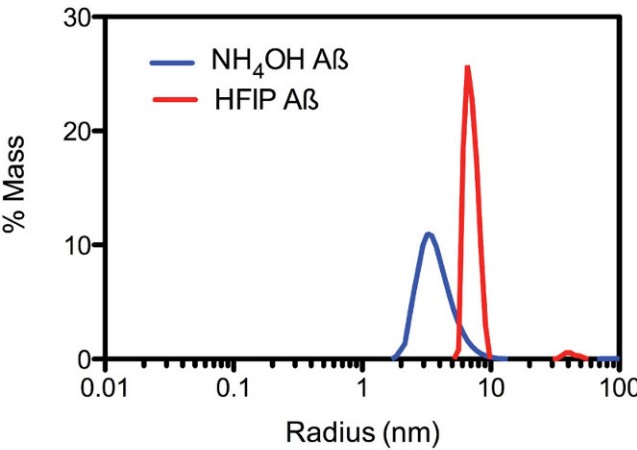

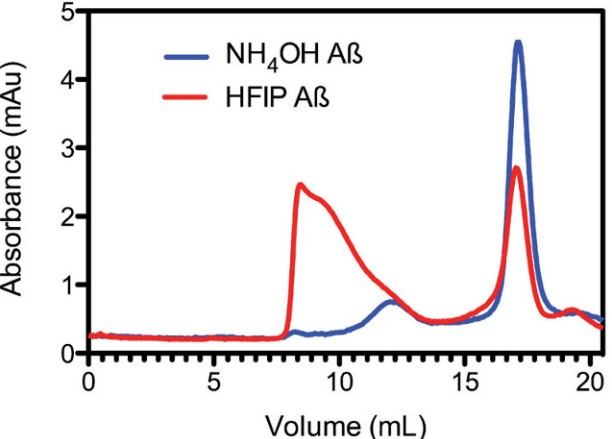

**Figure 2 Pretreatment method determines the starting aggregation state of A$\beta_{42}$.** (A) Hydrodynamic radii were determined with dynamic light scattering measurements of A$\beta_{42}$ (100 μM) after treatment with HFIP (red line) or NH$_4$OH (blue line). (B) Analysis of size distribution of HFIP (blue line) or NH$_4$OH (red line) treated A$\beta_{42}$ (200 μM) solutions by SEC using Superdex S200 media.

8 nm with approximately 50% polydispersity. In contrast, A$\beta_{42}$ pretreated with NH$_4$OH was more consistent, with the average distribution of five separate experiments showing radii ranging from 1–10 nm and 30% polydispersity factors. The average radius of this sample was 3.5 nm with 30% polydispersity (Fig. 2A).

Size exclusion analysis (Fig. 2B) of HFIP and NH$_4$OH pretreated A$\beta_{42}$ were consistent with the findings of the DLS experiments (Fig. 2A). We observed that the treatment of A$\beta_{42}$ with HFIP had an elution profile that displayed a heterogeneous mix of species, ranging in elution volume from 8.5 ml (the void volume of the column) to approximately 20 ml (the total volume of the column) indicating a large range of species. Using the area under the peaks to estimate the percentage of protein present, the major peaks from the HFIP pretreated peptide occur at 8.5 ml (∼65% total protein) and 17 ml (∼20% total protein) indicating the presence of monomeric (∼5,000 Da) and large oligomeric species

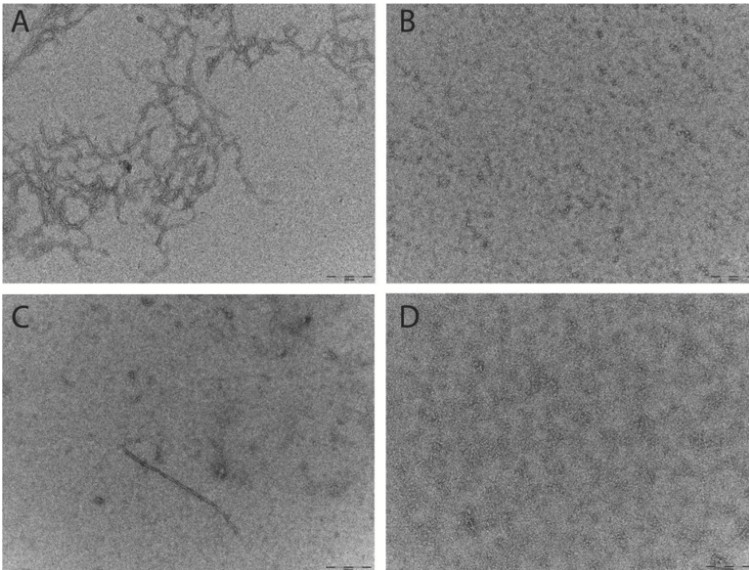

**Figure 3 Pretreatment method determines the starting aggregation state of A$\beta_{42}$ as determined by transmission electron microscopy.** (A & B) Micrographs from TEM analysis of HFIP (A) and NH$_4$OH (B) pretreated A$\beta_{42}$ as prepared for SEC and ThT, after 6 h of incubation. (C & D) Further micrographs were acquired for samples prepared for DLS pretreated with HFIP (C) or NH$_4$OH (D) Images captured 20 min post preparation. Scale bar = 200 nm.

(MW greater than 80,000 Da). For NH$_4$OH pretreated A$\beta_{42}$ we observed a different bimodal distribution with a major ($\sim$90% total protein) monomeric peak (MW $\sim$5,000 Da) eluting at 17 ml and a second minor ($\sim$10% total protein) peak eluting at 12 ml, corresponding to a medium sized oligomeric species (MW $\sim$40,000–50,000 Da).

## Electron microscopy of the preparation techniques

A$\beta$ prepared for the ThT and SEC assays were analysed by TEM. Micrographs show that after 6 h at 37°C the A$\beta_{42}$ pretreated with HFIP has more protofibril/early fibril structures compared with the A$\beta_{42}$ pretreated with NH$_4$OH (Figs. 3A and 3B). The smaller structures of the latter are beyond the sensitivity of this technique (Fig. 3B). In light of the differing buffer conditions used for DLS, the HFIP and NH$_4$OH pretreated A$\beta_{42}$ prepared for DLS were also analysed by TEM (Figs. 3C and 3D). The representative micrographs show that some short fibrils are present in the HFIP pretreated A$\beta_{42}$ solution (Fig. 3C), whilst only small, unidentifiable structures are present in the NH$_4$OH pretreated peptide solution (Fig. 3D).

## Small angle X-ray scattering of HFIP or NH$_4$OH pretreatment

In addition to the DLS and SEC measurements, we also investigated the initial aggregation state by SAXS measurements (Fig. 4). A comparison of the SAXS data of the untreated (Fig. 4A), HFIP treated (Fig. 4B) and NH$_4$OH treated (Fig. 4C) A$\beta_{42}$ highlights the different solution behaviour of these samples. The data indicates that all of the peptide solutions contained a polydisperse mix of aggregates, as the data in the low $q$ region of the scattering curves displays a significant increase in intensity, highlighted in

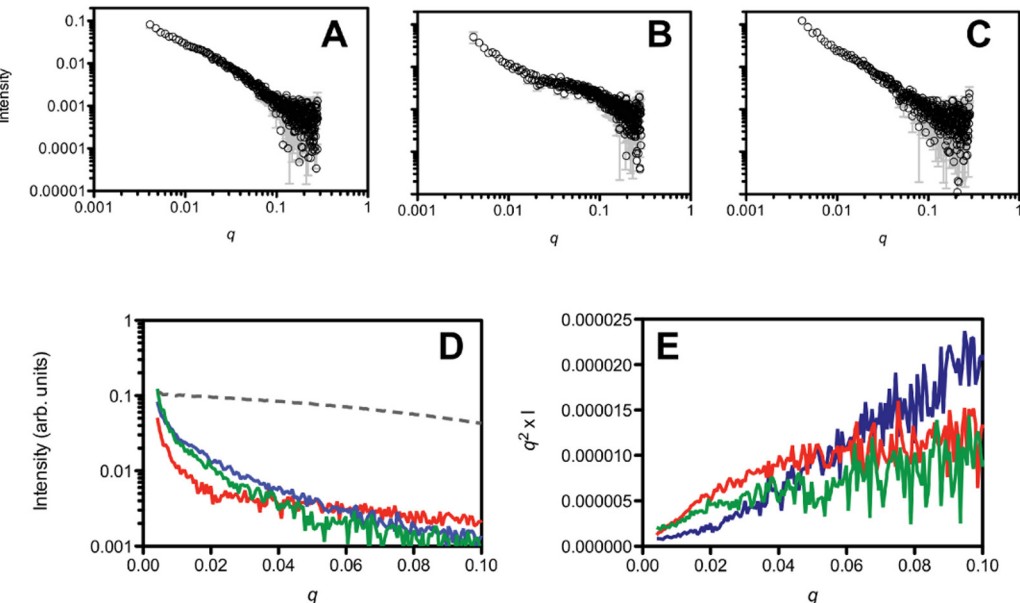

**Figure 4** **Small angle X-ray scattering analysis of the peptide treatments.** $NH_4OH$ treated (A), HFIP treated (B) and untreated (C) $A\beta_{42}$ (220 μM) solutions. Panel D shows a ln/linear plot of the data for untreated (green), $NH_4OH$ (blue) and HFIP (red) treated $A\beta$ over the range $0.001 > q > 0.1$. For comparison the scattering curve of the structured and non-aggregated ribonuclease A (dotted line) is also plotted. Panel E shows the corresponding Kratky plots of the data (untreated: green line, HFIP treated: red line and $NH_4OH$ treated: blue line).

Fig. 4D. This makes interpretation of the SAXS data in terms of shape or molecular weight difficult as the scattering from the aggregated material significantly distorts the analysis, however we can transform the data into a Kratky plot, which can give an indication of the folded/unfolded state of a protein (*Doniach, 2001*; *Mertens & Svergun, 2010*), with folded proteins typically showing a prominent peak in the curve, while unfolded proteins show a continuous increase as *s* increases (for review of the basis of Kratky plots and on the analysis of SAXS data see (*Doniach, 2001*; *Putnam et al., 2007*)). The treated and untreated $A\beta_{42}$ show a qualitative difference in the potential degree of folded peptide represented by the scattering curves (Fig. 4E). $NH_4OH$ pretreated $A\beta_{42}$ shows a completely unfolded structure, whilst HFIP pretreated peptide appears to represent partially folded species (Fig. 4E). The untreated peptide is intermediate in the degree of folded and unfolded peptide compared to HFIP and $NH_4OH$ treatment.

## The effect of pretreatment on cellular toxicity

Pretreatment of $A\beta_{42}$ has a large impact on the oligomerisation state of $A\beta_{42}$, which is a key factor for toxicity *in vitro*. A CCK-8 toxicity assay was used to investigate how the $A\beta_{42}$ preparation method affects toxicity in PC12 cultures after 24–96 h. Cultures treated with both $NH_4OH$ and HFIP pretreated $A\beta_{42}$ (15–20 μM) were viable after 48 h (Fig. 5A). Compared with vehicle treated cells, the PC12 cultures treated with $NH_4OH$ pretreated $A\beta_{42}$ (15 and 20 μM) for 72 h showed a 15.7% ($P < 0.001$) and 30.2% ($P < 0.001$) decrease in viability, respectively (Fig. 5B). In contrast, PC12 cultures treated with the

**A**

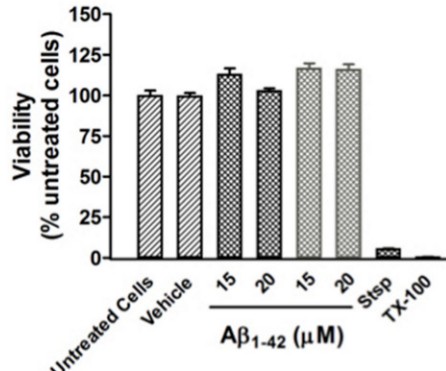

**B**

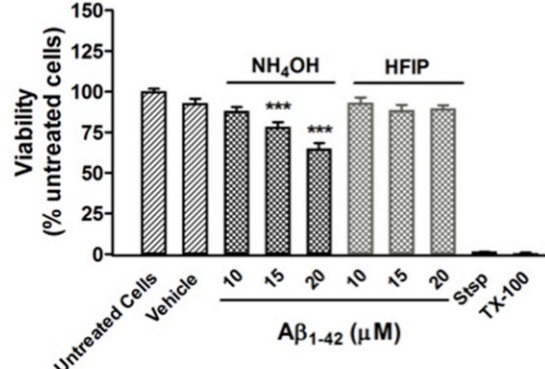

**C**

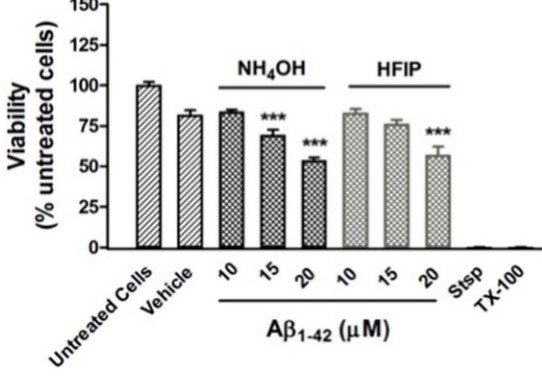

**Figure 5 Viability of PC12 cells assessed using the CCK-8 reagent.** (A) PC12 cell viability after treatment with NH$_4$OH- and HFIP-pretreated A$\beta_{42}$ (10–20 μM) for 48 h. 

**Figure 5 (...continued)**

(B) PC12 cell viability after treatment with $NH_4OH$- and HFIP-pretreated $A\beta_{42}$ (10–20 µM) for 72 h. (C) PC12 cell viability after treatment with $NH_4OH$- and HFIP pretreated $A\beta_{42}$ (10–20 µM) for 96 h. The toxin controls, staurosporine (Stsp, 1 µM) and Triton X-100 (TX-100, 0.1%), reduce the cell viability by $\geq$99%. Data represent the mean±SEM of 2–4 experiments performed in triplicate ($n = 6$–12). *** $p < 0.001$ for the $A\beta_{42}$ vs the vehicle treated cells. Grey bars = HFIP-pretreated $A\beta_{42}$.

HFIP pretreated $A\beta_{42}$ (10–20 µM) for 72 h did not show any loss of viability (Fig. 5B). However, the PC12 cultures treated with the HFIP pretreated $A\beta$ for 96 h, were 38.4% ($P < 0.001$) less viable than vehicle treated cells but only at the highest concentration tested (20 µM, Fig. 5C). After 96 h, the PC12 cultures treated with the $NH_4OH$ pretreated $A\beta$ (15 and 20 µM) showed a 25.2% ($P < 0.001$) and 42.1% ($P < 0.001$) decrease in viability, respectively compared with vehicle treated cells (Fig. 5C).

# DISCUSSION

Fluorinated alcohols such as HFIP are widely used to break beta sheet structure and induce alpha helical conformations (*Hirota, Mizuno & Goto, 1997*; *Buck, 1998*). These solvents have been used to inhibit processes involving beta sheet formation, mainly relating to amyloid aggregation (*Zagorski & Barrow, 1992*; *Buck, 1998*; *Stine et al., 2003*). The exact protocol for the use of these solvents appears to vary dramatically. However, most protocols describe an initial incubation with HFIP followed by evaporation to dryness and suspension of the resulting film with some other solvent. Common examples of the secondary solvent include DMSO (*Stine et al., 2003*; *Jiao et al., 2008*), ammonium hydroxide (*Yoda, Miura & Takeuchi, 2008*) or sodium hydroxide (*Ciccotosto et al., 2011*). In other cases the HFIP is not removed until after dissolution into aqueous buffer (for example *Kayed et al., 2004*), but this protocol is believed to be flawed as there is extensive evidence that low concentrations of HFIP are ineffective at dissociating oligomers (*Lesne et al., 2006*) and can actually enhance fibril formation (*Nichols et al., 2005a*; *Nichols et al., 2005b*; *Rangachari et al., 2006*).

Despite the use of HFIP for the purposes of erasing conformational memory, there is little understanding of whether or not disaggregation is maintained during the removal of the HFIP from the $A\beta$ sample and film formation. Previous work on $A\beta$ peptide film formation from HFIP solutions suggests that producing a film of the peptide results in accumulation of a variety of aggregated structures (*Pachahara et al., 2012*). Moreover, it has been suggested that HFIP disaggregation of $A\beta$ peptides is also inconsistent, with a DLS study showing that the peptide can self-associate within the HFIP solution (*Fulop et al., 2006*).

In this study, we have determined that HFIP pretreatment and film formation has adverse effects on the subsequent aggregation of $A\beta_{42}$ in an aqueous buffer system. Our study shows that HFIP treatment followed by sodium hydroxide resuspension adversely affects the starting aggregation state of the resuspended peptide, inducing a large proportion of small aggregates, otherwise unidentifiable by TEM but quantifiable by SEC, DLS and SAXS measurements. Interestingly, the starting aggregation state was

consistently worse with HFIP pretreated A$\beta_{42}$ independent of the secondary buffer used for biophysical analysis including both Tris.HCl pH 8.0 and PBS pH 7.4, indicating that buffer composition and pH 7.4–8.0 had little impact on the initial aggregation state. The large proportion of aggregate in the HFIP pre-treated A$\beta_{42}$ increases the kinetics of amyloid fibril formation, presumably due to increased number of nucli for fibril formation at $t = 0$. Similar to the observation of Stine et al.(*Stine et al., 2003*), the peptide films produced from HFIP treatment were difficult to resuspend consistently, leading to large variations in the concentration of the stock solution of peptide (data not shown). In contrast, treatment of A$\beta$ with NH$_4$OH results in fluffy white powder that yields consistent concentrations of A$\beta$ stock solutions (CV $\sim$ 10–15% based on the starting weight of the peptide).

Strong alkalis such as ammonium hydroxide, which do not approach the pI of A$\beta$ (pI $\sim$ 5) upon neutralisation, have been widely used as initial solubilisation agents for synthetic and *ex vivo* peptides. Solubilisation of amyloid by alkali has long been known to be effective, with the earliest report of its use in the isolation of amyloid from tissue samples occurring in 1898 (*Krakow, 1898*), followed by several other reports of the use of barium hydroxide, ammonium hydroxide, sodium hydroxide, calcium hydroxide, alkaline borate and alkaline buffered phosphate for solubilisation and analysis of amyloid deposits in tissue samples (*Shirahama & Cohen, 1967*; *Pras et al., 1969*; *Perry et al., 1981*; *Dubois et al., 1999*). In particular, sodium and ammonium hydroxides were observed to be the most effective in both preventing aggregation and disaggregating fibrillar structures (*Shirahama & Cohen, 1967*). More recently, ammonium hydroxide has been found to be of use in HPLC analysis of synthetic A$\beta$ peptides, where buffering the samples with 0.2% ammonium hydroxide results in improved peak shape, retention and sample recovery (*Lame, Chambers & Blatnik, 2011*). Furthermore, ammonium hydroxide pretreatment has been shown previously to be of use in producing homogenous solutions of A$\beta$ consisting primarily of low molecular weight species by Fezoui et al. (*Fezoui et al., 2000*; *Fezoui & Teplow, 2002*; *Teplow, 2006*). As initial pretreatment buffers, the aforementioned studies used 2 mM NaOH for A$\beta_{40}$ or 10 mM NH$_4$OH/6 M guanidine HCl for both A$\beta_{40}$ and A$\beta_{42}$ and the analytical techniques included fourier transform infrared spectroscopy, circular dichroism, atomic force microscopy, congo red binding, SEC, electron microscopy and neurotoxicity. Hence, our study is the first to directly compare HFIP to NH$_4$OH pretreated A$\beta_{42}$ using analytical techniques including ThT binding, DLS, SEC, SAXS, electron microscopy and neuronal cell viability. We found that ammonium hydroxide pretreatment and lyophilisation typically resulted in an A$\beta_{42}$ peptide residue that resembled "fairy floss". This lyophilised peptide resuspended extremely well when subjected to the sodium hydroxide resuspension protocol described in the methods section and the SAXS, DLS, SEC and electron microscopy analyses all indicated that the resulting solution contained a much lower proportion of structured aggregated material. The ThT analysis of the rate of aggregation indicated that the NH$_4$OH pretreated A$\beta_{42}$ gives consistent amyloid fibril formation kinetics, which were marginally slower then those of the untreated A$\beta_{42}$ peptide, suggesting that this protocol does not increase the amyloid forming

 

potential of the starting solution. This treatment protocol also decreased the variability in aggregation rate observed with the untreated peptide, with independent experiments producing differences in $t^1/_2$ of 5.2 h and 2.6 h for untreated and $NH_4OH$ treated peptide, respectively. In contrast to DMSO solvent, $NH_4OH$ has the advantage of being more compatible with analytical methods because low amounts of DMSO can interfere with UV absorbance measurements. In addition 0.5–1% DMSO is toxic to hippocampal neuronal cultures (*Hanslick et al., 2009*) and has some potentially adverse effects on A$\beta$ including denaturation (*Jackson & Mantsch, 1991*), inhibition of A$\beta$ aggregation (*Shen & Murphy, 1995*) and inhibition of A$\beta$ oxidative processes (*Davis et al., 2011*).

As observed by DLS, SEC and SAXS, the composition of the soluble material is different between the two pre-treatments (HFIP A$\beta_{42}$ has higher molecular weight aggregates compared with the $NH_4OH$ material). The HFIP pretreated A$\beta_{42}$ had a faster propensity to aggregate in the ThT assay (Fig. 1). The HFIP pretreated A$\beta_{42}$ was less toxic to PC12 cultures compared with $NH_4OH$ pretreated A$\beta_{42}$ at 72 and 96 h post-treatment (Fig. 5). These data suggest that the HFIP and $NH_4OH$ A$\beta_{42}$ may interact with the cell differently including differences in peptide affinity for the cell membrane, thereby affecting the toxicity. These data are consistent with the $NH_4OH$ material transitioning through a toxic intermediate, while the HFIP treated peptide appears to require extra time to develop toxicity. We hypothesise that the reduced toxicity of HFIP treated peptide is due to a decreased availability of effective (toxic) A$\beta_{42}$ because it is terminally trapped upon dissolution in a variety of non-toxic structures. Additionally, the observed delay in toxicity with HFIP pretreated A$\beta_{42}$ may be due to a requirement for disaggregation of already formed non-toxic aggregates to toxic oligomers.

The aggregation of the HFIP and $NH_4OH$ pretreated A$\beta_{42}$ peptide in the ThT and cell assays may be different due to the inherent complexity of cells and the effects of biological membranes, chaperones, metals and other proteins on aggregation kinetics of A$\beta$. There is strong evidence to support that once the A$\beta_{42}$ peptide is added to cell cultures, it binds to the cell membrane and is internalised (*Decker et al., 2010*; *Johnson et al., 2011*). Cell bound A$\beta$ structures range from monomers to hexamers and greater (*Johnson et al., 2011*). Whilst the A$\beta$ structure(s) which cause toxicity are unknown, there is a sequelae of pathological events including binding of A$\beta$ to synapse receptors (NMDA - (*De Felice et al., 2007*), $\alpha$7 Nicotinic acetylcholine receptor – (*Wang et al., 2000*)), an increase in calcium flux (*Demuro et al., 2005*; *Simakova & Arispe, 2007*), depolarisation of cell membrane (*Blanchard, Thomas & Ingram, 2002*), increased reactive oxygen species (*Parks et al., 2001*) and impaired axonal transport (*Wang et al., 2010*) which leads to cell toxicity. Thus, data interpretation across the *in vitro* cell-free and cell assays should be made with caution.

Using the preparation of A$\beta_{42}$ described in the present study, toxicity is observed at 72–96 h post-treatment with no toxicity at an earlier time point (48 h). The 72–96 h time frame for toxicity is consistent with that reported by others (*Fezoui et al., 2000*; *Barnham et al., 2003*; *Hung et al., 2008*; *Ciccotosto et al., 2011*; *Benilova, Karran & De Strooper, 2012*). Studies that show A$\beta$ toxicity in neuronal cultures after 1–24 h could reflect peptide with residual HFIP which is toxic (at concentrations >40 mM) to SH-SY5Y cells within the

24 h time (*Capone et al., 2009*). We hypothesize that the greater toxicity of $NH_4OH$ pretreated A$\beta$ at lower concentrations and earlier time points is due to a greater concentration of effective (toxic) A$\beta$. After HFIP treatment ~60% of the A$\beta$ were trapped as high molecular weight oligomers; provided this is also the case in cell experiments, these high molecular weight oligomers were not immediately toxic to PC12 cultures and thus additional time may be required for the high molecular weight oligomers to transition to the toxic intermediate. Alternatively, after HFIP treatment ~40% of the total A$\beta$ concentration were low molecular weight oligomers which may be too dilute to induce a toxic response in PC12 cultures. In general, these data indicate that the starting oligomeric status of A$\beta$ is an important factor for the observed toxicity.

In conclusion, we find that the use of HFIP alone is inappropriate for the stated purpose of producing a seed free solution of A$\beta$. We also find that ammonium hydroxide is a much more effective pretreatment which produces a mostly seed free solution of A$\beta$ upon resuspension with a sodium hydroxide protocol. We suggest that this approach is a much more appropriate treatment for biophysical and cell biology experiments of A$\beta$.

## ACKNOWLEDGEMENTS
SAXS measurements were acquired at the SAXS/WAXS beamline of the Australian Synchrotron (Clayton, Victoria, Australia).

### Funding
Funding was provided by CSIRO, a NHMRC program grant, and the Victorian government's operational support program. The funders had no role in study design, data collection and analysis, decision to publish, or preparation of the manuscript.

### Grant Disclosures
The following grant information was disclosed by the authors:
NHMRC program grant.
Victorian government's operational support program.

### Competing Interests
Colin L. Masters is an Academic Editor for PeerJ. Nigel Kirby and Haydyn D.T. Mertens are employees of SAXS/WAXS Beamline, Australian Synchrotron Company, Ltd. No other competing interests to declare.

### Author Contributions
- Timothy M. Ryan, Joanne Caine, Julie Nigro and Blaine R. Roberts conceived and designed the experiments, performed the experiments, analyzed the data, contributed reagents/materials/analysis tools, wrote the paper.
- Haydyn D.T. Mertens performed the experiments, analyzed the data, contributed reagents/materials/analysis tools.
- Nigel Kirby and Colin L. Masters contributed reagents/materials/analysis tools.

- Kerry Breheney and Lynne J. Waddington performed the experiments, contributed reagents/materials/analysis tools.
- Victor A. Streltsov analyzed the data, contributed reagents/materials/analysis tools.
- Cyril Curtain conceived and designed the experiments, analyzed the data, contributed reagents/materials/analysis tools.

## Supplemental Information

Supplemental information for this article can be found online at http://dx.doi.org/10.7717/peerj.73.

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
