# Peer review of "Ammonium hydroxide treatment of Aβ produces an aggregate free solution suitable for biophysical and cell culture characterization"

_PeerJ, doi:10.7717/peerj.73_

## Round 0.1 · original submission · Major Revisions

While two of the reviewers are generally satisfied with the paper and recommend publication after minor revision, a third reviewer is very critical and expresses some serious concerns about your results (although please note that lack of novelty is not a criteria for PeerJ). Any positive decision on the acceptance of your manuscript will be contingent on you and your colleagues addressing in detail the concern of this reviewer to whom any revision will be sent for further scrutiny.

Reviewer 1 ·

Basic reporting

The article is clearly written, provides sufficient background, and is generally well-referenced. I think they overlooked one important reference though: Fezoui, et al, Amyloid 2000, vol. 7, pp. 166-178, which I believe was the paper that established high pH pre-treatment as the method of choice to ensure that Abeta is disaggregated at the beginning of an experiment.

Experimental design

The goal of the research, i.e., to compare pre-treatment methods for ensuring Abeta disaggregation, is clear and the experiments are technically sound. However, as I am not an expert in SAXS I would have liked more information in the text on how to understand the Kratky plots. Also, while the authors were always careful to assure the reader that their final pH values for their assays were accurately reported, I would like more details on the buffer composition. In particular, what were the concentrations of phosphate and NaCl in the PBS? Also, the authors show a representative mass% vs. radius plot for their DLS results, but they state that the DLS results with HFIP pre-treated Abeta varied considerably. If possible (maybe as an inset?) could the authors show a broader sampling of their DLS data to give the reader a clearer idea of the extent of the variation? Along the same lines, while it is fine to show a representative trace for the thioflavin T kinetics, the authors should quote a mean +/- standard error for the half-times of fibril formation. Also, does the initial increase in thioflavin T fluorescence (within the first 3-4 hours) mean anything? Could it represent oligomer formation? If so, does pre-treatment affect this phase of the aggregation time course?

Validity of the findings

The authors’ results are very clear: HFIP pretreatment does not effectively disaggregate Abeta, and should be replaced by other methods, like high pH treatment.

Additional comments

No comments.

Reviewer 2 ·

Basic reporting

Bad grammar at places. Spell checking needed.

Experimental design

1)Importance of Scientific Question
The authors of the paper aim to demonstrate that the sole use of HFIP for the pre-treatment of aggregated A causes an “ increased proportion of oligomeric and aggregated material and an increased propensity to aggregate”. Moreover, they aim to show that an alternative method involving strong alkali pre-treatment results in “a much more homogenous solution that is largely monomeric”. This could be an important scientific question to address as many groups devise their own individual ways of preparing aggregate-free A samples.
2)Novelty
This paper aims to demonstrate that HFIP is not appropriate for producing aggregate-free A. However, this has been demonstrated and published previously by other groups, who have show that HFIP causing the formation of hydrogen bonds (Microsc. Res. Tech. 67,
164–174). The majority of papers published in the last 10 years use HFIP as a cosolvent in the disaggregation of amyloidogenic structures, and is followed by other steps including the addition of DMSO, strong acids, strong alkali or filtration. (Langmuir. 2010, 26(22):17260-8; Journal of Biological Chemistry, 272, 22364-22372.) Therefore, the novelty of this article is not particularly high. Many groups have published data demonstrating reproducible and reliable methods of producing aggregate-free A, using a range of protocols. This paper suggests the use of ammonium hydroxide as a way to disaggregate A. However, this is not a novel methods and has been used previously (BMC Biotechnology 2008, 8:82), and is also a method suggested by certain A production companies such as rPeptide. Therefore, the potential impact of this paper is relatively low. Even in the discussion they authors report that “Solubilisation of amyloid by alkali has long been known to be effective, with the earliest report of its use in the isolation of amyloid from tissue samples occurring in 1898”, therefore this provides no real new understanding towards A preparation or how HFIP influences assembly structurally.
3)Research Strategy
The method of A preparation does not seem the most suitable or consistent for producing a homogeneous peptide sample. Typically after solubilization, the peptide would be filter or centrifuged to remove pre-existing aggregates or contaminants. This is not performed in this study.
The authors use both DLS and THT to observe hydrodynamic radius and -structures respectively, however, they perform these measurements under significantly different conditions. The buffer solution, temperature and pH all significantly affect assembly of A. This is not mentioned in the manuscript or any suggest why such different conditions are used. So comparisons the reader wishes to draw are difficult.
Before performing the DLS measurements the authors syringe filter the A samples. Do the authors recheck the concentration of the peptide sample? as A adheres to surfaces and many filtration materials and this may significantly affect the concentration and aggregation propensity of the peptide samples.
The paragraph stating “Size exclusion analysis of HFIP and NH4OH treated A solutions were consistent with the findings of the DLS experiments (Figure 2B). We observed that the treatment of A with NH4OH had an elution profile that displayed a heterogeneous mix of species….” Makes no sense as the authors discuss the ammonium hydroxide preparation peptide twice and omit to discuss HFIP treated A.
The authors use DLS to measure the hydrodynamic radius of the peptide sample. However, the drawback of this measurement is that the hydrodynamic radius is only applicable to spherical particles. As A assembles to form a range of morphologies including small oligomers, paranuclei, fibrils and fibers DLS can cause problems with interpretation without complex data processing.
In the “effect of pretreatment on cellular toxicity” section the authors perform the toxicity studies using different A concentrations ranging between 10-20M. This would cause significant variability when trying to compare the pretreatment regimes. As reported previously by other groups, there is a marked difference in cell toxicity when using 1, 10 and 25M concentrations of A because of the assembly state of the peptide. Also, performing cell viability studies after 72 and 96 hrs seems like long incubation times when a significant proportion of the peptide would in fact be fibrillar, as even demonstrated by the authors here in their THT section. This could cause data interpretation difficulties.
Circular dichroism or other means of observing secondary structure of the peptides would be most beneficial. This would give a better understanding of initial secondary structure and in changes over time. Also, the addition of TEM or AFM would provide a good means of supporting the authors suggestion that HFIP-treated A is initially in an annular-state. (However, this has not been shown by other groups using similar HFIP methods of preparation, which have shown HFIP-treatment to produce small, soluble A samples).

Validity of the findings

4)Support of Conclusions by Data
The conclusions that HFIP pretreated peptide fibrillizes faster but causes decreased cell toxicity compared to NH4OH pretreated peptide does not seem viable. They suggest that the slower fibrillizing A caused by HFIP results from “dissolution in annular amyloid-like structures”. How would these annular structures dissolute? Annular amyloid would contain significant -sheet content, which is not observed in their THT data and dissolution of this annular material to form toxic A would suggest a decrease in THT signal which is not observed.
The THT data provides some concern as all three peptide samples appears to initially assemble immediately (increasing from 0-10 a.u). There doesn’t seem a true lag phase in any of the peptide samples.
The authors report that HFIP treated A does not cause cell toxicity at 10 or 20 M at 72hrs incubation and only causes cell toxicity at 20 M after 96 hours. It is well reported that HFIP-pretreated A causes significant cell death at concentrations lower than 10M within 5hrs.
From figure 2 the HFIP peak is much sharper and narrower suggesting a more homogeneous sample compared to NH4OH sample. Also, I would wish to see data extended beyond 100nm as it does not appear that 100% of the mass is represented in the graphs. Are there larger structures observable?
What is the cell viability of A-treated cells at 24hours? 72-96hrs seems long incubation periods to wait to observe cell toxicity at these peptide concentrations. Toxicity has been reportedly observed within 1-24hrs in other published data. Most of the peptide would be fibrillar at these time 72-96hrs.

Additional comments

The authors emphasize the work of reference 18 (Pachahara et al) as a motivation to demonstrate that HFIP induces the formation of amyloid structures in films of the peptide. Although, I agree with the very interesting paper published by Pachahara et al, I believe that the authors here maybe taking the work out of context. Pachahara et al demonstrate A forms -structures in HFIP in aged samples (2-4months) but do not show early time measurement. They also demonstrate the formation of -structures on mica, however, mica is hydrophilic and will also promote the formation of -structures. Therefore, in the context of this paper I do not believe this paper is appropriate unless they clearly state the relevance of Pachahara’s paper.
Figures 1 and 2. Keep the colors of NH4OH and HFIP consistent.
In the Aβ HFIP and “NH4OH pretreatment:” section. Was the complete removal of all NH4OH traces confirmed? e.g NMR, Mass spec. If traces of NH4OH are still present in the lyophilized sample this could significantly affect assembly of the peptide, delaying assembly and affect cell toxicity.
What are the sources of chemical and their purity. e.g TFA, HFIP
Concentration of buffer components not reported
Grammatical error throughout.
The authors begin to discuss formic acid solubilized A in the discussion, with no data shown. Either they should present the data in the results section, in supplementary material or omit it.
In the discussion the authors state they use pH7.4 during SAXS, DLS and SEC but report different pH in methods section
Concentrations of A in cell toxicity studies in methods and figure caption not consistent.
More suitable references should be added to show how other groups produce homogenously solubilized A.

Reviewer 3 ·

Basic reporting

OK

Experimental design

Should include TEM/AFM data to substantiate results presented in Figs 1 & 2.

Validity of the findings

The data will be robust and complete only if TEM/AFM data are also included.

Additional comments

The paper entitled "Hexafluoroisopropanol pretreatment enhances the amyloid
aggregation of Aß peptides in solution" presents interesting findings. However, characterization of aggregates described in Figs 1 & 2 need to be substantiated by TEM or AFM, particularly as sizes and shapes are part of the results and discussion.

---

## Round 0.2 · Minor Revisions

The revised version has been sent back to the more critical reviewer of the manuscript. After my own reading of your paper I came to the conclusion that his/her comments should be taken seriously into account before this paper can go to the publisher. I would particularly appreciate your comments on why different conditions were used for different biophysical experiments and why very high peptide concentrations were used for some of your experiments.

Reviewer 1 ·

Basic reporting

There is still a great deal of switching between American/British and British
spellings and missing words i.e lyophilised, lyophilization, solubilizing,
solubilisation. Needs to be read carefully

Experimental design

The biggest concern I have is that the authors state in the results section that they
hypothesize that “we hypothesised that this solvent was not completely removing
the oligomers, but was in fact seeding fibril formation.” This really concerns me
as could invalidate the whole paper without validation or confirmation. It could
easily be confirmed or excluded. HFIP should not be present in the peptide
solution as the authors point out that it itself is toxic to cells and in that case you
would expect the cells to die earlier than the NH4OH treated peptide if HFIP was
present in the peptide sample. As the authors state: In other cases the HFIP is
not removed until after dissolution into aqueous buffer (for example (Kayed et
al. 2004)), but this protocol is believed to be flawed as there is extensive
evidence that low concentrations of HFIP are ineffective at dissociating
oligomers (Lesne et al. 2006) and can actually enhance fibril formation
(Nichols et al. 2005; Rangachari et al. 2006), so this really needs to be addressed.

Validity of the findings

1) The authors state in the discussion that their suggested method is more compatible
compared to using DMSO as even low amounts can interfere, this needs to be
referenced and discussed further, many modern protocols ensure complete
removal of co-solvents.
2) I don’t think fibers can be described as “inert” structures as they can be a source
of monomers/oligomers through the continual dissociation/association events
between mono- oligomers and fibers.
3) In the SAXS methods section, the authors state they use a “slightly modified
approach to solubilizing” the peptide. All their methods from DLS, SEC, THT are
all different from each other anyway, however, if they state it is different could
they explain the reason why it is supposedly different from the rest?
4) At what concentration are the stock Aβ solutions prepared at for the THT
experiments? The authors say the working concentration is 5μM but don’t state
the stock conc. I only ask as they authors then state that the THT of the SAXS
method prepared Aβ samples has the same kinetics profiles and the SAXS
peptides are prepared at 1300μM which is a huge concentration to prepared Aβ.
5) Why are the peptides prepared for SAXS and cell culture at such a high
concentration? Is there a reason, as this seems incredibly high peptide stocks
conc?
6) I appreciate that the authors state in their response that it is difficult to compare
the data between the tht, dls, saxs, cell culture because they use different
preparation conditions such as pH, buffer composition but this still concerns me
as ideally the reader would like to be able to compare the results in each technique
as, I the reader, would like to be able to compare the most toxic cell culture time
point with what the tht data points out etc. I am still confused as to why all these
techniques can not be performed under the same conditions much like other
groups do that have published papers comparing in vitro and in vivo results.
7) The authors suggest that the HFIP pretreated peptide results in increased rate of
fibrillization because the presence of solvent. This can more than easily be
verified or exclude this possibility by checking for the presence of HFIP in the
sample. NMR could easily be used to confirm the presence of HFIP, (or another
suitable technique).
8) In the DLS results section the authors suggest that the HFIP treated peptide has
polydispersity of 50% as the peptide ranges from 3-100nm and that NH4OH has
only a range of 1-10nm but this is confusing in the Supplementary figure ½. Are
we the reader looking at the black average line or the low/high values because
HFIP goes up to 1000nm if we look at the high values and the NH4OH goes up to
200nm so it’s a little confusing what we should focus on.
9) As a reader I am still concerned about the cell toxicity assay, under these
preparation conditions and concentrations you would really expect to see toxicity
at a much earlier time point that 72-96hrs. Could the authors reference other
papers to show comparable results to their findings?
10) The authors hypothesis that the HFIP treated peptide is trapped in a variety of
non-toxic oligomers using this pretreatment of HFIP however it has been
suggested by other publications that immediately solubilized peptide using this
method results in immediate permeation of membranes via calcein release assay
and in vivo. And if the authors believe that there is still HFIP in their peptide
sample this could cause their observed in vivo toxicity results.
11) If the authors hypothesize that the HFIP treatment causes other cellular events to
occur, what do the authors suggest is happening? Is it the presence of HFIP in the
sample or the effects HFIP has on the peptide structure and aggregation sate?

Additional comments

In the discussion section the authors mention “film formation has adverse effects”.
This could be a very interesting avenue to explore in the paper as none of the
current results in this paper confirm that the film formation is causing aggregation.
Various biophysical techniques could be employed to examine the secondary
structure of the peptide in the dry film.

---

## Round 0.3 · accepted · Accept

I found you response to the reviewer's comments comprehensive and satisfactory.